# Improving best practice for patients receiving hospital discharge letters: a realist review

Katharine Weetman,[1] Geoff Wong,[2] Emma Scott,[1] Eilidh MacKenzie,[3] Stephanie Schnurr,[4] Jeremy Dale[1]

¹Unit of Academic Primary Care, Warwick Medical School, University of Warwick, Coventry, UK
²Nuffield Department of Primary Care Health Sciences, Medical Sciences Division, University of Oxford, Oxford, UK
³Warwick Medical School, University of Warwick, Coventry, UK
⁴Centre for Applied Linguistics, University of Warwick, Coventry, UK

**Correspondence to**
Katharine Weetman;
K.Weetman@warwick.ac.uk

## ABSTRACT

**Objective** To understand how different outcomes are achieved from adult patients receiving hospital discharge letters from inpatient and outpatient settings.

**Design** Realist review conducted in six main steps: (1) development of initial theory, (2) searching, (3) screening and selection, (4) data extraction and analysis, (5) data synthesis and (6) programme theory (PT) refinement.

**Eligibility criteria** Documents reporting evidence that met criteria for relevance to the PT. Documents relating solely to mental health or children aged <18 years were excluded.

**Analysis** Data were extracted and analysed using a realist logic of analysis. Texts were coded for concepts relating to context, mechanism, outcome configurations (CMOCs) for the intervention of patients receiving discharge letters. All outcomes were considered. Based on evidence and our judgement, CMOCs were labelled 'positive' or 'negative' in order to clearly distinguish between contexts where the intervention does and does not work.

**Results** 3113 documents were screened and 103 were included. Stakeholders contributed to refining the PT in step 6. The final PT included 48 CMOCs for how outcomes are affected by patients receiving discharge letters. 'Patient choice' emerged as a key influencer to the success (or not) of the intervention. Important contexts were identified for both 'positive' CMOCs (eg, no new information in letter) and 'negative' CMOCs (eg, letter sent without verifying patient contact details). Two key findings were that patient understanding is possibly greater than clinicians perceive, and that patients tend to express strong preference for receiving letters. Clinician concerns emerged as a barrier to wider sharing of discharge letters with patients, which may need to be addressed through organisational policies and direction.

**Conclusions** This review forms a starting point for explaining outcomes associated with whether or not patients receive discharge letters. It suggests several ways in which current processes might be modified to support improved practice and patient experience.

## INTRODUCTION
### Background

*Discharge communication* may follow an inpatient or outpatient discharge; it typically comprises written discharge information in the form of a discharge letter or summary. It

### Strengths and limitations of this study

► First study to review and develop realist theories about patients receiving discharge letters.
► The engagement of patients, general practitioners and commissioners in refining the programme theory increased relevance and rigour of the theory.
► The programme theory is likely to be applicable and relevant to multiple healthcare settings.
► The exclusion criteria imposed restrictions on the programme theory such that evidence relating to children, solely to mental health and those lacking capacity is not considered.
► Only sources written in the English Language were included.

is a well-established practice that the physician who is to follow-up patient care, typically the general practitioner (GP) or equivalent,[1] should receive written *discharge communication* from the discharging physician; this practice supports continuity of care between specialist services and primary care. Patients are sometimes included in this communication, and while within the UK this is considered to be 'good practice',[2] is not standardised.

The *Department of Health* in the UK describes patient copies of letters as a 'right'[3] and recommend patients should be copied in where appropriate as a 'rule', unless there is risk of harm.[2 3] This is intended to support patient understanding and well-being, increase patient safety and the quality of information sent and improve doctor–patient relationships.[2–4] More recently, the *Academy of Medical Royal Colleges* released the 'please write to me'[5] initiative. The initiative encourages doctors to write directly to patients in simple plain English to increase understanding. Despite these initiatives and guidelines, evidence within and outside the UK reports both benefits[6–10] (eg, patient satisfaction), and drawbacks[11–15] (eg, patient confusion) of patients receiving their letters.

While patients receive copies of discharge letters inconsistently,[16 17] the reasons for this and the subsequent consequences remain unclear.

Hence, the objectives of the current study were to conduct a realist review of patients receiving discharge communication (the intervention); to develop a programme theory (PT); and to make best practice recommendations. The research questions (RQs) were as follows:

RQ1: What positive and negative outcomes have been reported on patients receiving written discharge communication?

RQ2: What are the important contexts which are associated with whether the mechanisms produce the different outcomes, and why?

## METHODS

A realist review is a, 'theory-driven, interpretative approach to the synthesis of evidence'.[18] Synthesising evidence involves interrogating data sources to develop, refine and test *context, mechanism* and *outcome* configurations (CMOCs). 'Context' may be conceptualised as external factors that influence mechanisms.[19] 'Mechanisms' are hidden, context-sensitive causal forces that produce 'outcomes'.[19] Following Pawson *et al*,[19–22] CMOCs should be configured and consolidated to build and develop a realist *PT* or theorised explanation of how an intervention *works* or not. The intervention under scrutiny 'patients receiving discharge letters' was defined by the review team as 'the patient being given or sent any form of written (paper or digital) hospital discharge communication; this could be a direct copy, patient-directed letter, or a combination.' The aim of the review is to understand and explain how the different outcomes are produced for adult patients receiving written discharge letters. Outcomes may be simplified into desired/beneficial or 'positive' (eg, increased satisfaction) and undesired/detrimental or 'negative' (eg, increased anxiety).

We have previously published the full protocol for this review[23] which justifies the rationale for a realist approach and considers each of the methodological steps in detail. The overall review design was informed by previous literature, driven by the RQs, consists of six steps[19 24 25] and is further described in the protocol paper.[23] This design is summarised in figure 1.

### PT development (step 1)

The task of locating existing theories to develop an initial rough PT was achieved through a scoping search. Theories and evidence were sought which aided explanation of how and why patients receiving discharge communication results in different positive effects (eg, drug adherence) and negative effects (eg, preventable hospital readmissions). Sources were selected based on their 'relevance'[19–21] to the PT; where *relevance* concerns 'does the [source] address the theory under test?'.[20] Crucially, the whole source did not need to inform the PT but we considered the relevance and contribution of sections of the document.[20]

Search terms were based on the intervention (eg, patient cop(y)ies). Published resources and healthcare websites were searched to ascertain a range of evidence (see online supplementary file 1). During this phase, research team judgement was needed to decide the stopping point for PT development as was the need to balance the degree of comprehensiveness and practicalities.[23] As the purpose was to locate existing theories and initial concepts, the search was not intended to be comprehensive and the decision was made to screen no more than 30 documents. During the scoping search, search strategies within articles and article indexing were noted in order to inform a more thorough subsequent search in step 2.

Twenty-seven documents were selected from the scoping search (see online supplementary file 2). All documents were then interrogated and coded for any CMOCs, concepts or theories which could inform development of a PT. These were consolidated to form figure 2, the initial PT.

The initial PT shows two main channels for discharge communication; patient copied into (or not) the hospital to GP letter and patient received a personalised letter. Limited evidence was available for the option of 'patient does not receive copy' as evident in figure 2. Patients being copied into discharge letters, whether by choice or otherwise, are associated with a large range of mechanisms and

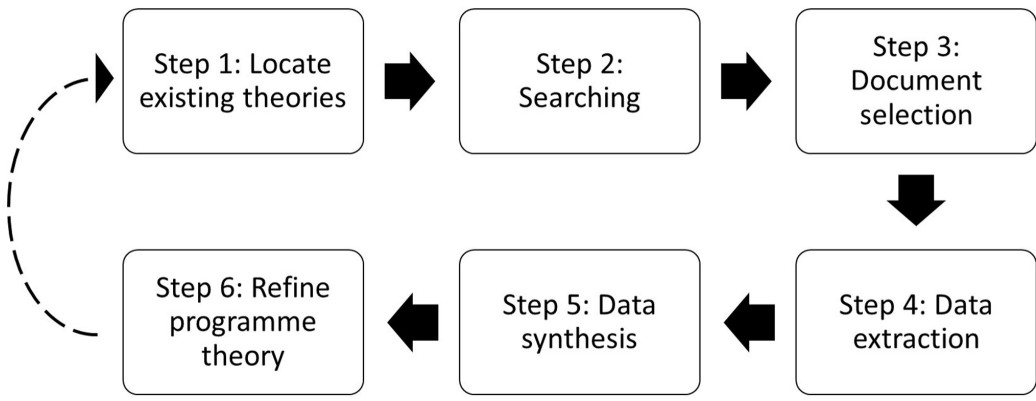

**Figure 1** Review design.

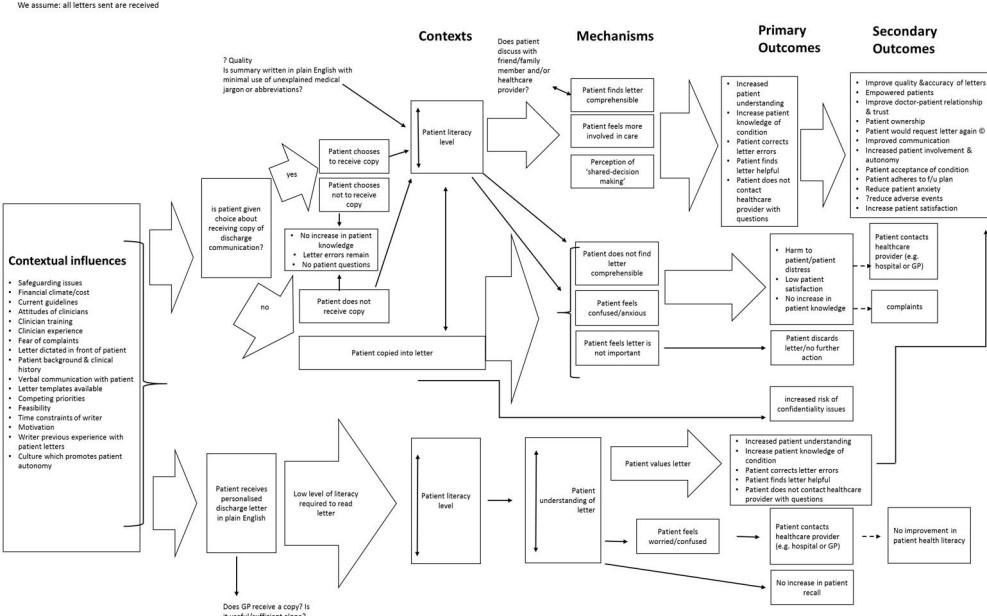

**Figure 2** Initial programme theory. GP, general practitioner.

outcomes. Contexts such as 'patient literacy level' are likely to influence generation of mechanisms (patient does/does not find letter comprehensible) and outcomes (increase/no increase in patient knowledge), but this was unclear from the evidence reviewed in the scoping search. The scoping search revealed a range of 'contextual influences' (eg, 'time constraints of writer' may affect whether a patient is given a choice about receiving a letter and also the overall letter quality). It was unclear where some CMOCs began and ended (eg, 'is patient given choice about receiving copy of discharge communication' falls between contextual influences and context labels). In figure 2, f/u stands for 'follow-up' and the small circled 'c' icon refers to an outcome which could also thereafter take the form of a context. In summary, there were clear 'gaps' and information missing from the initial PT, confirming that the scoping search in isolation was insufficient for realist theory generation; further evidence and searching was needed to clarify details and simplify CMOCs (steps 2–6).

## Search strategy (step 2)

The electronic searching was purposive and guided by the initial PT, results and indexing from step 1. A search strategy was developed which was piloted and adapted for MEDLINE until a diverse and relevant range of search results were yielded (target 500–3000). In line with a realist approach, searching was iterative, and the strategy was refined for each database (see online supplementary file 3). Sources included electronic databases, healthcare sites and grey literature.

The search strategy was not intended to be exhaustive, but provided a large enough overview to be meaningful for PT development.[20] Evidence was searched up until September 2017; publications were monitored thereafter but no new evidence affected the PT. In total, 3113 documents were selected for screening.

## Selection and appraisal of documents (step 3)

Inclusion or exclusion of source evidence for the review was according to the following criteria:

### Inclusion criteria
► Full text or section of source had *relevance*[19 20] to informing the PT.
► Relate to inpatients/outpatients discharged from general hospital setting to GP (or equivalent).
► Relate to discharge where 'written discharge communication' is sent to GP or referring physician (may also be copied to patient).
► Source written or published in English.

### Exclusion criteria
► Specific to discharge to units/physicians other than GPs (or equivalent), for example, another hospital.
► Specific to discharge of patients who lack cognitive capacity, for example, dementia, or where there may be higher risk of harm, for example, mental health discharge.
► Lack of written communication having taken place, for example, telephone only.
► Specifically relate to patients <18 years.
► Source not written or published in English.

The exclusion criteria posed limitations on the review; children under 18 (where the parent would often be the letter recipient), patients with particularly specialised communicative needs (eg, patients without capacity) or where the intervention may have a higher potential risk of causing harm (eg, psychiatric discharge documents) were excluded. The communication needs of

some of these patients may be more complex and variable within and between groups and therefore was not possible within review scope. The first exclusion criterion states patient discharge communication to those other than GPs or equivalent (eg, family or community physicians) was excluded. This is because the review specifically focussed on discharge communication to GPs and patients rather than referrals or care-handovers. Furthermore, the review aimed to develop a theory for patients receiving discharge communication and inclusion of hospital–hospital discharge may have reduced clarity and produced a less focussed theory.

Once KW had screened the documents by title and abstract, second reviewer EM screened a random 10% test selection; this proportion was selected following Wong *et al*.[24] Inter-reviewer agreement was set at kappa measure κ≥0.8.[26] A result κ<0.8 would require all documents to be second screened. Inter-reviewer agreement was calculated as sufficient (κ=0.82). In the first screening phase, 611 duplicates were removed and 2341 documents excluded; this left 161 documents.

The full texts of these 161 documents were then screened, primarily for relevance[19 20] by KW, with EM screening a random 10% sample. Inter-reviewer agreement was again sufficient (κ=0.92). Eighty-eight documents were excluded at this stage leaving 73 for inclusion.

In addition, hand-searching of bibliographies, 'cited by' searching, and contacting experts was undertaken. This identified a further 30 relevant documents, creating a total of 103 documents. Online supplementary file 4 provides the final document list. The selection process is summarised in figure 3.

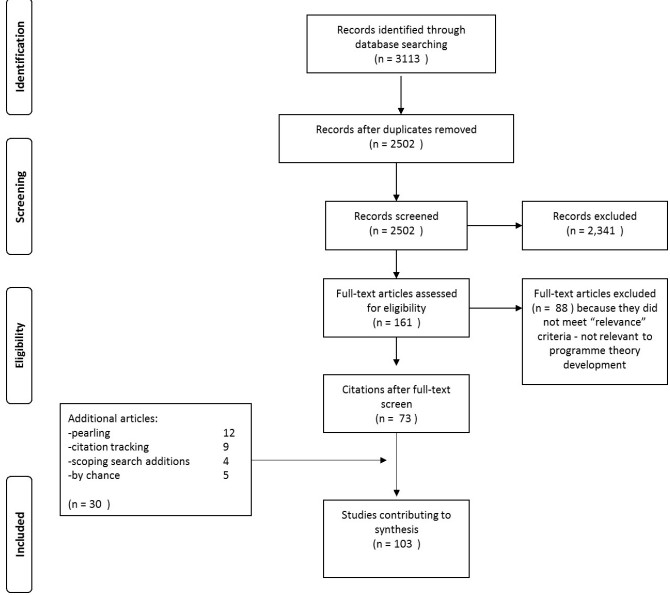

**Figure 3** PRISMA[111] diagram (document selection process). PRISMA, preferred reporting items for systematic reviews and meta-analyses.

### Data extraction and analysis (step 4)

A hybrid approach to data extraction was undertaken.[24 27 28] This allowed extraction of both descriptive document characteristics and annotation of CMOC ideas for synthesis and integration into the PT.[19 20] A data extraction form was designed iteratively to record pertinent document details. Final columns included: author(s), year, geographical information, healthcare system, design aim, number of participants, intervention, clinical specialty, inclusion and exclusion criteria, findings/conclusions, rigour/quality assessment,[19 20] topic focus, form of discharge communication, for example, discharge summary, participant mix, staff mix and relevance score.[19–21]

Documents were also annotated in NVivo for CMOCs and PT ideas. Annotations were guided by the initial PT devised in step 1.

### Data synthesis (step 5)

During step 5, data and annotations of PT ideas and CMOCs were consolidated. A realist analytic approach, following the work of Pawson *et al*,[19–21 29] was used to interrogate the theory during data synthesis. Pawson *et al*[19–21 29] present several different frameworks for synthesising data evidence. We selected the framework[20] entitled 'synthesis to consider the same theory in comparative settings', which involves five analytical strategy steps. This framework was chosen as it assumes theories sometimes 'work' and 'do not work' according to the particular setting; Pawson *et al*[20] describe this as 'aim[ing] to make sense of the patterns of winners and losers'. Hence, this framework is suitable for the RQs which focus on cause and context of positive outcomes 'winners' and negative outcomes 'losers'. Thus, data synthesis was grounded on the assumption that the *outcomes* of the intervention may differ according to *context*.

The following realist analytical strategy steps[20 27 30] were undertaken simultaneously:
1. Juxtaposition of data sources—align sources to build on/clarify each other.
2. Reconciliation of data discrepancies—explore reasons for data disparities.
3. Adjudication of data—data quality consideration of trustworthiness/relevance.
4. Consolidation of data—inference of mechanisms for outcomes.
5. Situation of evidence—consideration of intervention settings.

Data synthesis using the analytic strategy 'juxtaposition of data sources' was achieved through utilisation of NVivo 'nodes'. Sections of text were annotated, and coded as nodes. The nodes were named according to ideas or concepts around the PT and contained sections of text that were used to build CMOCs. NVivo node coding resulted in 19 nodes as seen in table 1.

During, 'reconciliation of data discrepancies'[19 20 24] and 'adjudication of data',[19 20 24] the data coded within NVivo was used for scanning and comparing data to

## Table 1 Coding nodes

| Node name | No of different sources coded | Total no of sections of text coded |
|---|---|---|
| Autonomy | 5 | 5 |
| Clinician context (views) | 23 | 57 |
| Confidentiality | 12 | 15 |
| Context (when it does not work) | 29 | 46 |
| Context (when it does work) | 54 | 107 |
| Cost/resources | 20 | 33 |
| Dictate in front of patient | 3 | 5 |
| Doctor–patient relationship | 5 | 7 |
| GP preference | 4 | 8 |
| NHS policy or contextual standards (international) | 30 | 51 |
| Outcomes (positive) | 58 | 128 |
| Outcomes (negative) | 22 | 28 |
| Patient as delivery method | 2 | 2 |
| Patient harm | 24 | 33 |
| Patient letters | 18 | 34 |
| Patient preference | 37 | 94 |
| Patient recall | 11 | 12 |
| Queries and contact | 10 | 12 |
| Understanding | 46 | 88 |

GP, general practitioner; NHS, National Health Service.

identify disparities. Adjudicating and situating evidence was important to reconcile discrepancies.[19 20 24] We interpreted the data coded within each node and judgements were formed as to which sections of text might be functioning as contexts, mechanisms or outcomes. We then made assessments about what the CMOC might plausibly be for each CMO based on the data within each node. Where relevant, we also drew on data contained within other nodes to build CMOCs. Following this process, a CMOC table was constructed (see online supplementary file 5) for consolidation of data.

After table completion, following Pawson *et al*'s framework,[20] it was important to make sense of the 'winners' and 'losers'. CMOCs were primarily labelled according to how evidence was reported in the included documents, such as whether the outcomes were described as desirable or beneficial. Where evidence was limited or outcomes were not clearly described or evaluated, the research team interpreted what data were available and formed judgements about these CMOCs based on content expertise in order to generate 'positive' and 'negative' labels. CMOCs were not limited to one per document or one per patient experience. Thus, multiple outcomes and CMOCs could be annotated for a single experience; this exemplifies the complexity of the intervention under scrutiny.

Notably, there were a greater number of CMOCs relating to positive outcomes than when the intervention does not work.

### PT refinement (step 6): patient and public involvement
Review step 6 was to consider stakeholder perspectives to test and refine the PT in light of the synthesised data[20] and to assess whether the PT aligns with real-life experiences.[18] We invited comment from local policy makers and health service commissioners, GPs and a patient and public participation group. Groups were selected according to convenience and accessibility through University links. They were invited to suggest refinements to the PT in an entirely voluntary format, and all comments were anonymised. Formal ethical approval was not required[23] but informed involvement was sought.

### RESULTS
### Document characteristics
The 103 evidence sources were from 16 countries across various continents with most emanating from England (54%), the USA (17%) and Australia (7%). Healthcare settings were split between insurance style systems (23%) and publically funded systems (77%), such as the National Health Service (NHS). The date range of the sources was from 1979 to 2017 and the total number of participants detailed across the research studies was 16 383; this included staff and patient participants but there was not enough detail across all of the studies to quantify the participant type proportions. Most had been published in the 10 years prior to the search: 1970–1979 (1%), 1980–1989 (2%), 1990–1999 (7%), 2000–2009 (40%) and 2010–2017 (50%). The source type was mixed: discussion and opinion pieces (20%), survey-based study (19%), guideline documents (12%), abstracts (7%), review (5%), interview-based study (5%), experimental study (5%), pilot study (5%), randomised controlled trial or randomised intervention study (5%), non-randomised intervention study (3%), report document (3%), cohort study (2%), mixed methods (not covered above) (1%) and other, for example, PhD thesis (8%).

The evidence covered a wide range of specialties. Most specified inclusion of adult patients only (over 18 years) but often did not detail the exact patient ages in the write up; a few studies focussed on elderly patients. Information relating to patient demographics, for example, gender, was often not found in the sources and hence these were not summarised. Many sources instead focussed on the specialty under consideration in the document and clinical presentations of interest to that specialty for example, ECGs.[31] Participants who were staff included medical students, hospital doctors of all training grades, nurses, GPs, non-specified hospital staff and non-clinical staff. However, the majority of documents (66%) either did not provide staff participant details or they were irrelevant, for example, guideline document, no participants. The type of discharge communication varied: direct copies

(47%), discharge instructions (12%), pictures (1%), personal discharge packs (1%), personalised letters (12%), information booklets (9%), multiple types of discharge communication (7%) and other (11%). Where the sources came from showed some variation such as Department of Health archive (3%) and conference listing (5%) but the greatest number of sources were from journals (68%).

### Quality and document rigour
The findings were considered in light of the quality of included documents. During data extraction, documents were quality appraised for *rigour* and evaluated for *relevance*.[19 20] The concept of *rigour* is defined as 'whether the methods used to generate the relevant data are credible and trustworthy'.[18] *Relevance* and *rigour* were scored on a scale from very low to very high and factors such as document type (eg, opinion piece or scientific trial paper) were considered. Documents were not excluded solely based on rigour as extracts of documents with a lower quality score may still have valid contributions.[20] The quality of evidence varied, with 53% of sources graded as medium or above for relevance and 80% for rigour. Information relating to setting and context was often limited.

### Context–mechanism–outcome configurations
The following section provides an overview of theories in the form of a narrative of how patients receiving discharge letters does or does not work, as informed by the evidence reviewed. The subheaded themes emerged during data interrogation and consolidation although many acted as 'nodes' in earlier annotation and coding (see table 1). Sections contain references to CMOCs, quotations from data texts and references. Quotations have been chosen which illustrate the described theories and highlight key elements of CMOCs. The full table of 48 CMOCs is found in online supplementary file 5.

Evidence relating to some aspects of the PT was limited, particularly in relation to negative outcomes, intervention costs, current clinician views, impact on doctor–patient relationships, personalised patient letters and patients not receiving any intervention. Evidence was also thin in relation to data disparities. Although context and outcome information was generally well supported, mechanisms were frequently omitted. Where possible, based on the evidence and research team expertise, we inferred reasons for disparities and what the likely mechanism(s) were within any CMOC.

### Patient preference/choice
Allowing patients to make their own choice for receiving letters may reduce unnecessary resource strain[32][CMOC14], only take minimal time,[9] make patients feel more involved in their care[2 9 33–37] [CMOC2], increase satisfaction[10 16 34 38–42] [CMOC14, CMOC41, CMOC47] and aid information acceptance[14]:

> I wanted to know as much as possible about what was going on with my body.[14] (p.73)

Sometimes for whatever reason you don't fully take on board what the doctor has told you. I found the letter useful to read over and digest properly what was written.[2] (p.3)

Many patients report that receiving letters is useful.[2 7 16 32 36 37 39 43 44] Patients may show friends/family to help them better understand their condition/treatment.[14 16 34 36 42 45–47] Patients may use letters as a reference/reminder for the consultation[14 16 34 36 39 40][CMOC45]:

> My mind went blank after seeing the doctor and the letter reminded me of what had been said.[36] (p.83)

Across a range of specialties and settings, the reported patient preference for receiving copies of their discharge letter is generally high (79%–97%).[7 8 14 34 35 40 43 44 48–51] However, not all patients may find letters helpful[32 42] or necessary,[7 32–34 39 42 52] and some may not want to be reminded of their diagnosis,[34] which could decrease satisfaction, and generate queries if these patients were sent letters without a choice [CMOC40]. Hence, several studies argue in favour of respecting patient choice and suggest that the patient's right to 'opt out' needs to be addressed[7 14 16 39 49 52–54] [CMOC41]. In situations where the patient is not offered a choice, such as third-party information or risk of harm,[2] the clinician should be able to justify this decision.[55] In relation to sensitive information or social diseases, patients generally do not object to this being included in the letter as long as it has 'some relevance'.[55]

### Comprehension, queries and recall
There was considerable evidence, particularly from patient viewpoints, to support the view that the majority of patients may understand their letters[7 8 14 15 32–36 43–45 55 56] and hence find the letter beneficial and helpful[32 43 57] [CMOC7, CMOC39, CMOC44]. Moreover, a letter copy which is understood can reassure patients that they are being listened to[42 43 58] [CMOC34, CMOC48]. Patient understanding of discharge instructions may increase their knowledge[42] and this might reduce unnecessary or avoidable hospital readmissions[59–63] and reduce patient anxiety[64] thereby supporting patient well-being[16 50 55] [CMOC39]:

> I found the letter very comforting and reassuring.[65] (p.58)

Although there may be a risk that patients receiving letters is associated with an increase in queries to seek clarification about what has been communicated,[54] several sources indicated that this occurs to a minimal extent[8 33 43 51 55] [CMOC29]. Examples of patients not understanding letters were sometimes described as a 'small proportion'[32] or low percentage '7%'.[34]

If patients are provided verbal information only, they may fail to retain the information[36 40] [CMOC18] which can decrease recall and adherence [CMOC21]:

By the time I have got home, I have forgotten half of what was said in clinic.[43] (p.255)

Due to this, combining written and verbal information[66] may improve patient understanding[36 67–70] [CMOC15, CMOC18], increase patient's involvement in their care[16 36] and compliance[2 17 44 58 71 72] [CMOC11, CMOC43] and improve recall[10 14 15 33 40 44 68 73 74] [CMOC5, CMOC15].

Contexts which may increase likelihood of triggering patient understanding include letter language translation,[38 75 76] writing the letter at a fifth-grade or sixth-grade reading age level[38 68 77 78] [CMOC12], use of glossary,[16 79] pictures, pictographs or equivalent[68 79–81] (particularly for low literacy or illiterate patients) [CMOC17], lay explanations for 'technical terms'[16 55] and writing in plain English with minimal jargon/abbreviations[2 35 39 55 68 78 82–84] [CMOC12].

Two sources with the same lead author suggested training medical students on writing patient letters can help produce letters that are more meaningful to patients[65 85] [CMOC13]. However, the evidence around training in relation to the intervention was limited and needs further research.

### Personalised or patient-directed discharge letters

Producing a letter which is comprehensible and useful to both GPs and patients has been recognised to be an issue.[17 47 77 86] Patient-directed or personalised patient letters have been proposed [CMOC24, CMOC36]. Patients often rate these letters positively[6 42 87] and this may heighten satisfaction,[42] and improve understanding[6 88]:

> Simplifying written communication has also been shown to improve patient comprehension.[6] (p.855)

However, personalised letters have the potential to lead to resource consumption[44] [CMOC25], staff time depletion[32 44 88] and patient anxiety that they have been given different information to their GP[88] [CMOC26]. For these reasons, further research which weighs the benefits of personalised patient letters against the drawbacks and costs is needed.

### Patient to deliver letter

The context of patients delivering letters seems to have few reported positive outcomes. Posting and electronic transferral of letters may be preferable as:

> It is not considered good practice to send the discharge summary home with the patient as there is no guarantee that the information will be passed on to the general practitioner.[89] (p.7) [CMOC31]

### Dictating letters in front of patients

Evidence for this concept was somewhat thin. One study suggested that dictating letters in front of patients can make patients feel less in need of a copy of the letter.[8] Another paper suggested this practice may also provide a context that triggers patients to challenge inaccuracies, improving letter quality[90] [CMOC22, CMOC30]:

> The content of letters to GPs is sometimes incorrect and this may be remedied by dictating the letter in front of the patient.[90]

### Confidentiality

There are concerns and legal implications surrounding potential confidentiality breaches associated with patients receiving letters, particularly when they are sent out in the post.[35 40 48 50 53 86 91] One recent paper[48], which looked at confidentiality, continued to stress risks around postal communication and the importance of secure information transfer:

> There is a substantial risk of breaching patient confidentiality when distributing correspondence by post. A well-designed security arrangement is therefore required to ensure the safety of confidential information.[48] (p.35)

Some documents[2 35 48 53 55] suggested ways to reduce potential risk of confidentiality breach through communication platforms and the processes involved in sending letters, for example, verifying patient contact details before sending letter[35 58] [CMOC3, CMOC27, CMOC28].

### Patient harm

Patient anxiety or 'harm' in general are often cited as reasons for clinicians not wanting to copy letters, particularly in 'bad news' settings[14–17 36 44 47 56 92] [CMOC6]. Letter inaccuracies can cause concern leaving patients feeling confused or anxious[32] [CMOC19]. Nevertheless, the letter can reassure the patients their problems are being handled[50] [CMOC46] and initial anxiety can settle or be nullified by the usefulness of the letter[7 41–44 55 85 91] [CMOC37]. Moreover, one study[39] published in the *Lancet* suggested that patient letters in 'bad news' settings may be more useful than 'good news':

> Patients who had received bad news found the letter significantly more useful in helping them to understand and remember what they had been told during the consultation than did patients receiving good news… almost half the patients receiving bad news found their letter distressing to some extent; however, with 1 exception, all patients were pleased to have received it.[39] (p.924–925)

Although the above paper was published in 1991, we found no recent evidence or system changes to dispute the notion that 'bad news' letters may be of particular use to the patient. Hence, despite risk of initial 'harm', 'bad news' letters should perhaps not be avoided.

Practical and feasible suggestions were found in some documents for minimising harm or anxiety: not copying letters with information not previously disclosed to the patient[2 3 14 55] [CMOC38], abstain from use of value judgements, for example, pleasant female[36] [CMOC12],

potentially avoid or carefully consider copying letters where there are 'problems of privacy at home' and/or where the patient lacks capacity[2] [CMOC20], and checking the patient consents to a letter[55] [CMOC41].

## Clinician views

GP and hospital clinician views were described both as broadly in favour[9 33 47 58 88 93 94] [CMOC5, CMOC16] and not in favour of patients receiving written discharge communication across a range of specialities[10 11 16 33 35 36 45 47 50 88 94 95] [CMOC6, CMOC35]. The response section[9 11 86] to a *BMJ* article[96] on patient letters demonstrates the clinician view dichotomy as practitioners argue for and against patients receiving letters:

> My colleagues and I have had to explain to alarmed and bewildered patients who have received copies of their correspondence the meaning of phrases…[86] (p.1369)

> The purposes of clinic letters are to communicate with general practitioners and keep a legible record in the notes of what is happening and what might happen. It is written in medical speak, and it is fantasy to suggest that letters written like that will ever be meaningful, without further explanation, to most patients.[11] (p.1369)

> Generally, doctors who are sceptical about copying letters to patients seem not to have tried it, whereas those who send copies routinely are enthusiastic.[9] (p.1370)

Practitioner perceived benefits found in the sources [CMOC5] included improved patient understanding,[47 88] increased transparency[45] [CMOC33], improved trust/doctor–patient relationship,[9 47 88] dispelling fears of 'secretive relationships' between clinicians[47] and heightened sense of patient importance.[47] In addition, the patients' right to view the information was noted[88] [CMOC7, CMOC16]. A common practitioner concern of the intervention across specialties was letter comprehensibility and patient understanding[11 16 33 35 36 45 47 50 86 88 97 98] [CMOC6]. Other concerns included cost of additional materials/staff time[17 32 33 48 50 86 88] [CMOC23, CMOC32], patient anxiety[16 17 35 36 47 50 88 95] [CMOC6, CMOC19], increased patient queries[17 33] [CMOC29], potential confidentiality breaches[47] [CMOC6, CMOC27] and that letters would need to be oversimplified.[16 17 47 88 98 99] An attitudinal issue found in two oncology documents[45 88] published 17 years' apart was the view that letters are tools to be used between doctors only [CMOC6]. Additionally, juniors can learn from and mimic superiors and also not send letters to patients.[95]

Confusion around 'letter comprehensibility' and lack of 'patient understanding' were the the most common clinician reservations relating to the intervention.[11 16 33 35 36 45 47 50 86 88 97 98] However, as covered in the *comprehension* section, patients are often reported as *understanding* their letters[7 8 14 15 32–36 43–45 55 56] and furthermore they tend to express strong preference for receiving such letters.[7 8 14 34 35 40 43 44 48–51] Thus, it may be inferred from the evidence that patient understanding of letters is possibly higher than clinicians' perceive.[33 44 56] The following from a recent abstract[45] concisely summarises an example of patient and clinician view disparity:

> While some oncologists assess the copy letters as inappropriate for supplemental patient-oncologist-communication, breast cancer patients regard this tool as predominantly gainful. Oncologists appear to stick to their traditional perspective which perceives the copy letter mainly as a communication tool from doctor to doctor.[45] (p.185)

Notably, much of the evidence reporting clinician views was published from 2002 to 2008 and current evidence on clinician perspectives remains limited. Moreover, although sources occasionally referred to conflicting clinician views, information on why attitudes differ was thin. Overall, better understanding of current clinician views on copying discharge letters to patients is required. Further research should address reasons behind different viewpoints to include patients and practitioners.

## Cost and resources

The estimated costs associated with the intervention varied[16] but this must be considered in the context that included documents spanned a wide time range and thus factors such as inflation need to be considered. In addition, robust health economic analyses were not found in the included sources. Documents[16 17 32–34 36 40 43 49–51 53–55 86 90 100] referred to 'cost' or financial implications [CMOC25] of sending letters in different ways such as use of consumables[17 32 33 49–51 54 100] [CMOC10], and secretarial[16 17 32 33 36 43 50 54 55 100] [CMOC10] and clinician time required.[17 32] A few sources,[2 17 34 36 44 55 101] including guideline documents and research papers, suggested that benefits were such that associated costs were minimal, or even reduced by patients being more informed from receiving discharging communication [CMOC7, CMOC25, CMOC42]. However, as many of these views were based on personal comment or studies with weak methodologies, the true cost consequences remain unknown.

## Autonomy

One source suggested that when patients are not given letters, they may feel less involved in their care, resulting in reduced patient autonomy[41] [CMOC1, CMOC6]:

> …to refuse to provide such information if this is the patient's wish is to deny their autonomy.[41] (p.388)

Conversely, some evidence[16 41 102] was found that providing patients with written discharge letters is their 'right',[3 53] may create a sense of involvement, and increase patient autonomy and satisfaction [CMOC2, CMOC4, CMOC5, CMOC8, CMOC14].

## Doctor–patient relationships

Few documents[2 9 14 16 47 72 88] were found which considered the intervention in terms of the doctor–patient relationship. However, much of the limited evidence that was found indicated that patients receiving letters has the potential to improve communication, trust and the doctor–patient relationship [CMOC9].[2 14 16 47 72]

## Stakeholder perspectives

As detailed in step 6, the final review step was to refine the PT through stakeholder perspectives. Three groups were consulted: local commissioners, GPs and service-users/patients. Stakeholder involvement took the form of group discussions and email correspondence. As the PT was continually being developed throughout the review process, stakeholders commented on the most recently developed PT at the time of their involvement. Groups were relatively small; due to feasibility, it was not possible to achieve diverse and representative group samples.

Group discussions were centred on the PT; members were encouraged to critique and feedback on the PT diagram. This included concepts not covered or explored in detail in the PT diagram such as the importance of comprehensible language and terminology, difficulty and problems retaining verbal information only for example, following use of anaesthesia, patient choice of receiving letters, illegibility of handwritten discharge communication, critical context of prior patient communication of a high quality to increase likelihood of understanding discharge letters, issues around personalised patient letters considering NHS resource availability, and concerns around writing a letter which meets the needs of both GP and patient. The commissioner and GP representatives emphasised the importance of patient safety and that this should be central to best practice recommendations. In addition, the patient group reported reading a letter about themselves written in third person was peculiar. The patient group felt patient letters were very important for patients taking responsibility for their

health in line with the NHS promotion of patient-centred and patient-led care.

Several different members across the various groups commented that in practice, patients do not always receive their letters, despite this process being recognised as best practice. Recommendations were suggested to rectify this by the commissioner members to include the following: clinicians should assume when writing letters that they could be made available to the patient, early clinician and student training in good letter writing and record keeping and that hospitals should support the initiative for example, quality improvement activities and audits.

## Cycling of review steps

As a realist review is an iterative process, steps may be repeated. As described in step 2, new publications were followed and consulted for evidence but provided no new or conflicting PT knowledge. Thus, it was deemed that 'theoretical saturation'[19 21] in accordance with Pawson's realist review methodology[19–22] was attained and no further searching or step cycling was required.

## Resultant PT

The PT was systematically updated to produce a resultant PT following review steps 1–6 (figure 4). This still shows two main channels for CMOCs: patient copies of letters and patient personalised letters. There remained limited CMOCs for where patients do not receive letters, due to the paucity of evidence available. Contexts for when the patient does receive their letter(s) were condensed into an aligned grouping of five key contexts for when the intervention may be theorised to work and four key contexts for when the intervention may be theorised not to work. The feasibility of providing a personal patient letter was updated on the PT; findings from both stakeholder feedback and data synthesis suggested personalised letters may currently be more feasible in private or insurance-based healthcare settings than in the NHS. In addition, the box of contextual influences was deleted,

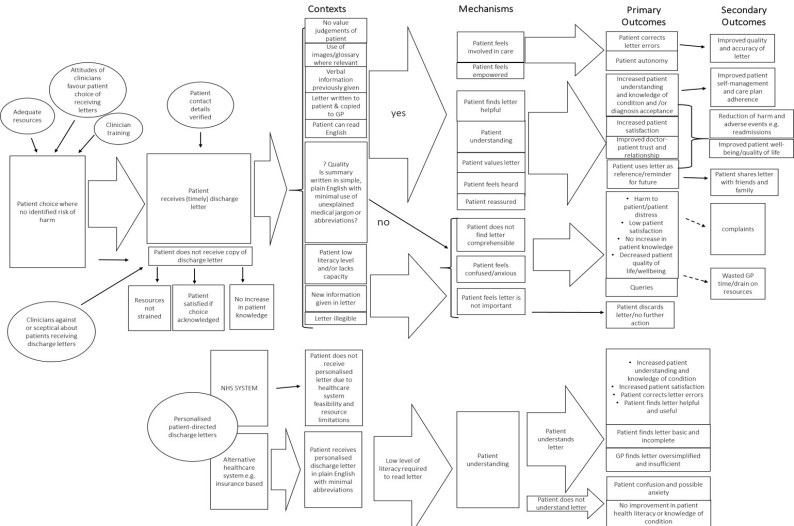

**Figure 4** Resultant programme theory. GP, general practitioner; NHS, National Health Service.

and the points integrated into the overall diagram. Patient outcomes throughout the PT were simplified and clarified (eg, the outcomes such as 'empowered patients' and 'reduce patient anxiety' were simplified to the outcome 'improved patient well-being').

In addition, CMOCs have been 'grouped' where overlap was apparent, for example, all resources are labelled simply as 'resources' as data often concurrently referred to financial, administrative and clinician time resources. 'Patient choice' emerges as a key influencer to the success (or not) of the intervention. Notably, there are a range of contexts, mechanisms and outcomes for when the intervention is theorised to 'work' (eg, positive outcome=improve doctor–patient relationship), and when it does 'not work' (eg, negative outcome=harm to patient). It is also evident that contexts can be used in combination to increase the likelihood of beneficial outcomes; this is indicated through wide arrows to show multiple linkages and amalgamations (eg, a letter could contain no value judgements of patient and be written in simple plain English). Notably, the resultant PT has a higher incidence of CMOCs for when the intervention 'does work'; this is considered further in the discussion.

## DISCUSSION
### Statement of principal findings
This review of 103 sources summarises and expands on existing evidence by moving beyond 'benefits' and 'drawbacks' of patients receiving letters alone, and considering contexts of *when* as well as *how* the intervention works. Although the review focuses on the UK health system, our use of realist review has enabled identification of findings that may be transferable to other healthcare settings.

RQ1 asked about positive and negative outcomes of the intervention. Positive outcomes include increased patient satisfaction,[10 16 34 38–42] improved doctor–patient relationship and trust,[14 16] heightened patient knowledge,[42] improved letter and record quality,[2 55 90] and reduced anxiety.[64] Negative outcomes include patient queries,[54] confusion[47 50] and anxiety.[15 42]

RQ2 enquired after the important contexts for triggering these outcomes. Important contexts for positive outcomes include letters written in plain English with minimal abbreviations,[55] lay explanations or simplified terms in brackets for medical jargon,[16 55 82] for example, myocardial infarction (heart attack), written information provided alongside verbal explanation,[66] no new information in letter[2 14] or value judgements,[36] letter translation[38 75 76] where relevant, training clinicians on letter writing practice,[2 65 83 85] use of pictures and glossaries where relevant,[16 68 79–81] letters only given to patients who choose to have them,[16 32 55] and where there is no identified risk of harm[2] or confidentiality breach.[55] Important contexts for negative outcomes include patient choice not acknowledged,[55] only verbal information provided,[43] letter involving terms and a style that is too advanced for patient to comprehend,[77] and letter sent without verifying patient contact details.[35 53 58]

This review has produced two key findings, which are important but not wholly surprising. The first is that the reviewed evidence indicates that patients value their discharge letter and their understanding of them is possibly greater than clinicians perceive.[33 44 56] However, reasons behind patient and clinician perceived comprehension discrepancies were unclear. It is important to situate the first finding in terms of the study exclusion criteria and participant diversity across the evidence reviewed, for example, it is likely that patients who participate in research on this topic have a greater level of interest and literacy than those who did not participate. One or a number of demographic groups not involved in the studies, either by choice not to participate or by exclusion, may have accounted for a portion of those who clinicians perceive to have low understanding. Thus, evidence for low patient understanding was limited, and this requires further research. The second key finding is that in a number of contexts, patients expressed preference for receiving correspondence.[7 8 14 34 35 40 43 44 48–51] Patients can continue to use the letter(s) to refer to beyond discharge,[16 34 36] as a medication list reminder, and to share with friends/relatives as desired.[15 34 36 88] Nevertheless, patient *choice* should still be acknowledged as the review did find evidence that not *all* patients want their letters; a practical way of addressing this would be to check with the patient that they want a letter in the first instance.[7 16 53 55]

Systems for monitoring patient letters (eg, the Newcastle Trust Policy for auditing and sharing letters with patients[55]) seem prudent moving forward. This is of particular relevance in the NHS given that guidelines for copying letters have been widely available since 2003[2] and yet in practice, many patients do not receive letters.[16 48] Given the wider context of a drive for patient-led care and patient-centred communication and decision-making,[17 41 103] this review is timely and relevant. The review findings have the potential to influence policy and improve practice. The results demonstrate how care can be improved through patient choice and good quality letter provision. However, current clinician views[33 35 36 44 45 56 88] and hierarchical mimicking of practices of seniors[95] pose a barrier to implementation and need addressing.

### Review limitations
For this review, we followed the Realist And MEta-narrative Evidence Syntheses: Evolving Standards (RAMESES) quality and publication standards for realist reviews.[104 105] Quality assessment and analysis is to a degree dependent on reviewer skills and reflexivity.[106 107] Furthermore, analysis and inferences were 'subjective and interpretative'.[108 109] However, because the steps we have taken for this review are transparent, other review teams can see and make judgements on result plausibility.

Due to lack of time, it was not possible to involve all stakeholder groups who may be connected with discharge

communication. Nonetheless, the review had a specific focus on 'receiving' discharge communication and thus stakeholders were targeted who were closely associated or involved in policy of discharge letter receipt.

The resultant PT is limited by the quality and content of evidence reviewed. Some of the evidence found in sources was markedly thin, particularly in relation to costing information, recent clinician viewpoints, personalised letter copies and influence on the doctor–patient relationship. Furthermore, there were a greater number of CMOCs relating to positive outcomes than negative outcomes, that is, when the intervention *does* work than *does not*. This may be rationalised by publication bias towards positive findings. Additionally, the binary distinctions between positive and negative outcomes, that is, when the intervention does and does not work, may have imposed oversimplified CMOC labels. 'Positive' and 'negative' labels were based on evidence presented in the documents reviewed; at times a degree of subjectivity was involved in this process. Although these binary labels (positive/negative) may have oversimplified some CMOCs, we felt the usefulness of clear distinctions between when the intervention was interpreted to 'work' (and not) outweighed the drawbacks of this method.

CMOCs for patients *not* receiving letters (nil intervention) were thin. Consequently, these evidence limitations constrained the detail available in the resultant PT in these areas. Additionally, not all mechanisms could be inferred from the data resulting in some visible mechanism 'gaps' within the CMOC table (see online supplementary file 5).

The review and PT are not exhaustive but this is not the intention of a realist review.[110] Given time and resource constraints, the review was limited to adult patients who had been discharged from general hospital settings, and other patient groups were excluded. Furthermore, the PT is limited by the representativeness and diversity of the patient groups within the sources reviewed. Much of the evidence was drawn from small scale studies conducted in single settings, and even within these there is likely to have been participation bias which will have resulted in the views of ethnic minorities, patients lacking literacy and other marginalised groups being under-represented.

### Suggestions for future research

The PT offers a useful starting point for future research and should be useful and practicable for informing policy and guidelines. Further research is needed to explore the relevance of the PT to groups, such as children and adults being discharged from mental health services, who were excluded from this review and to those, such as marginalised groups, who may have been under-represented in the evidence included in the review. There is also a need for research to define the cost benefits of copying patients into discharge letters in order that the importance of this topic and the consequences of poor practice are recognised by policy-makers, managers and professional bodies. Potential barriers such as clinician

views and the current limited available clinician training on letter writing should be addressed; research and evaluation is needed to inform how this can be effectively achieved. Since patient and clinician views were sometimes conflicting, a study which parallels both views alongside the same patient cases to understand reasons for any discrepancies would be useful and may provide valuable insights. This is the topic of a PhD that is currently being undertaken by the lead author, and will be reported in due course.

### CONCLUSION

The resultant PT forms a basis for explaining how, when, why and for whom this intervention does and does not work. The resultant PT makes suggestions for how best practice of patients receiving discharge letters may be improved to enhance the provision of patient-centred care. Evidence for some aspects of the PT was rather limited, indicating a need for more research.

The key findings are that the value patients place on discharge letters and their understanding of the letters' content is possibly greater than clinicians' perceive, patient choice is instrumental to increasing the likelihood of desired outcomes and that clinician views may act as a barrier to wider practice implementation. This could be addressed through clinician training and organisational initiatives which guide, mandate and monitor the intervention. Without such organisational support, it is unlikely that current processes will be consistently improved given the barriers identified in the review.

In conclusion, this review describes how the intervention of patients receiving their discharge letters may *work* to increase the likelihood of positive effects and reduce potential negative effects.

**Acknowledgements** We thank Samantha Johnson, our information specialist, for critically appraising the search strategy. We would also like to thank all stakeholders involved in the review for improving the quality and relevance of the resultant theory.

**Contributors** KW was the lead reviewer. EM completed the second reviewer tasks. All authors contributed towards the programme theory through discussions. KW is responsible for the design and drafting of the initial manuscript. GW, ES, SS and JD critically reviewed and edited the final manuscript. All authors read and approved the final manuscript.

**Funding** This work is supported by the Economic and Social Research Council (ESRC) and Clinical Commissioning Groups (CCGs) of Coventry and Rugby and South Warwickshire. Funding for the open access charges for publication of this review was provided by the RCUK fund.

**Competing interests** None declared.

**Patient consent for publication** Not required.

**Provenance and peer review** Not commissioned; externally peer reviewed.

**Data sharing statement** This review presents previously published and publicly available data. Please refer to the reference list (in article and in supplementary file 4) and their authors for these research data.

and indication of whether changes were made. See: https://creativecommons.org/licenses/by/4.0/.

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
