## [Reviewer comments · BMJ Open]

ARTICLE DETAILS

TITLE (PROVISIONAL)	Improving best practice for patients receiving hospital discharge letters: a realist review
AUTHORS	Weetman, Katharine; Wong, Geoffrey; Scott, Emma; MacKenzie, Eilidh; Schnurr, Stephanie; Dale, Jeremy

VERSION 1 – REVIEW

REVIEWER	Jean Macq UCLouvain, Belgium
REVIEW RETURNED	15-Nov-2018

GENERAL COMMENTS	Thanks for inviting me to review that interesting article. It presents very well the whole process of doing a good realist review. Methodology is globally well explained and how this translate into results is also thoroughly detailed. As a consequence of those detail, it gives difficulty for the reader difficulty to “grasp” the main aspects of the theory, and the findings leading to it (i.e. one has to wait the conclusion to understand key points). So, in order to make the paper more readable, 1) I would suggest to make “synthetic” points at different stage. Namely the initial PT and the updated PT lack of “clarity at this stage. On P4 line 49 the initial PT is presented on fig 2. It may be usefull to explain synthetically the content of that figure. Indeed, as it is, it is already very “complicated” (a lot of content). As a consequence, the reader has difficulties in the getting the “essence” of it, and the rationale of the initial PT. The same reader will therefore come with series of questions in the choices done in designing the initial PT (for example, why providers characteristics, interactions between GPs and hospital providers, ... is not mentioned in the context, and not taken in the mechanisms?). On p15. As for the fig 2, it would increase visibility to explain synthetically the content of fig 4 2) I would suggest to reduce details of some of the contents (for example, simplify the explanation on quality of evidence p9) Few detailed comments P3 line 53. Intervention is explained but not outcome. I would insert the explanation given on p4 line 16 just after the explanation of the intervention. P4 line 26. “what are the important contexts”.. add characteristics / components?... “which are associated..” P7 line 45. Typo error? The word Intervention repeated twice
---

REVIEWER	Prof Maggie Bartlett University of Dundee School of Medicine, Scotland, UK
REVIEW RETURNED	11-Dec-2018

GENERAL COMMENTS	This is a well conceived, well planned and well written paper which follows the protocol for a realist review. The background is adequately explained. The methods section is very detailed and includes most of the necessary information, though the question of ethics approval needs to be addressed. Patients and NHS clinicians were involved and I would have expected the inclusion of a statement regarding the decisions about the need for ethics approval and their outcomes. The results section is clearly written and presented. The diagrams of the Programme Theory and its subsequent revision are comprehensive and easy to follow.
--

REVIEWER	Sarah Yardley Marie Curie Research Department, University College Hospital, United Kingdom
REVIEW RETURNED	22-Dec-2018

GENERAL COMMENTS	This is a really impressive and well conducted piece of work which only suffers from a few points of clarification and the need to more fully explain and justify elements of the methods and limitations. The authors should be congratulated on what they have achieved and with minor amendments I consider it to be a great addition to the literature. My suggestions are as follows: a) I accept the authors' point that benefit and harms that might arise from patients receiving or not receiving discharge letters has not been well researched or evidenced to date. However, it is quite hard to follow their logic in the abstract / introduction / methods (much clearer in the results but the reader needs early orientation) regarding the implications of this observation for the current review study as to how lack of understanding played out with respect to what outcomes were sought or not (including evidence about 'absence of outcomes' when constructing their programme theory. It concerns me however that as presented the study initially appears to have only looked for positive outcomes (see methods para 2 re the protocol) and how these could be reproduced rather than also looking for negative outcomes and/or unanticipated / unintended consequences (which might be either negative or positive). Later under review design negative outcomes are mentioned but it is still not clear if and how these would be accounted for in the analysis. The research questions then clearly state both negative and positive will be sought (RQ1) and contexts for both will be sought (RQ2) but again refer to whether mechanisms will produce different outcomes without specifying if this is all possible outcomes (as how defined e.g. in literature, by patients / stakeholders / by authors) or only those perceived as positive (and by who). Aside from referencing the protocol I think it is important this is discussed more clearly in the current submission, particularly as outcomes will not be mutually exclusive / binary in nature (a patient may experience 2 positive and 1 negative outcome for example) and it may be there is a programme theory to be developed for further work investigating if the counter to CMOs for negative outcomes would in fact be beneficial. b) I was surprised to see that only general practitioners and health service commissioners were included as stakeholders. The
--

	authors should consider the implications of not including hospital professionals who write discharge letters for both their findings and the potential impact of their study further in the limitations section. c) Exclusion reasons  1. Why was mental health excluded? This needs better justification because if anything there is a history in Mental Health of greater transparency (not least due to the MHA as well as the MCA) and harm is often overestimated by professionals 2. Why were children excluded? While the letter would go to the patient's parents / guardian's for what reason did the authors anticipate this to be so different as to exclude? 3. It would have been, in my view, more appropriate to have included all letters related to discharge to the patient's home or usual place of residence e.g. other community teams rather than just GPs. Can the authors comment further on these three decisions? In all instances above I'm unconvinced by the justification that needs of these groups would be more complex and variable - surely this is a reason to include and not to make assumptions - inclusion would have helped produce a more robust theory so this should be discussed as a limitation. In a small way not doing this reinforces the view of some healthcare professionals that their area of practice is a special case without evidence to support such a view. d) I strongly advice the editors that table 6 is moved from supplementary data to the main article as this will strengthen the impact.
--	---

REVIEWER	Natalie Forster Northumbria University, UK
REVIEW RETURNED	10-Jan-2019

GENERAL COMMENTS	This paper makes an important and original contribution to current evidence by examining how, why copying patients into hospital discharge letters leads to different outcomes, and the key contexts which facilitate the success of this practice. The realist approach is well justified and appropriate to study aims. The programme theory developed gives a comprehensive picture of implementation pathways and outcomes (including potential unintended outcomes) of patient discharge communication. The review also draws out a number of useful suggestions for practice and further research on this topic. However, I do feel that greater clarity could be provided on some aspects of study methods, particularly in regard to the processes of developing and refining programme theory. Specific comments are as follows: While the abstract gives an accurate summary of the paper, it would be helpful to add detail of the outcomes of receiving discharge letters that are under consideration in here. The large amount of literature synthesised forms a key strength of the review. However, I felt the methods section would benefit from a more precise account of what the different review stages entailed, and how these fed into the development of initial
--

	programme theory. This would enhance transparency in regard to how findings were produced. It is suggested that stage 1 involved a scoping search to locate existing theories, which were then used to develop an initial programme theory. It would be helpful to state what type of theories and evidence sources that were sought in this phase e.g. was this a search for broader abstract theory, or for existing articulations of programme theories in terms of how discharge letters are supposed to work? It is recognised that judgement is required in regard to the stopping point for developing programme theory, however it would be helpful to say how and when this judgement was reached, especially since this can be a common challenge when using realist approaches. How were decisions made to include articles in the scoping stage from among those available? A clearer distinction could also be made between the scoping search for existing theories (step 1) and electronic searches of the literature (step 2). The search terms/approaches seem very similar for these steps, yet within realist approaches, different search methods are often used depending on the level of theory being sought and developed (see for example Andrew Booth's chapter in Doing Realist Research, 2018). A discussion of how these two stages differed from one another and their relative contributions to the development and refinement of the programme theory would be helpful. At times I found it difficult to follow the reporting of the synthesis process. I feel that a stronger thread is needed to show how the analysis progressed through the different stages and how these fed into one another e.g. how exactly did the authors move from annotations on individual articles (according to CMOC and PT ideas) to nodes? And then from the nodes to the CMOC table? E.g. did authors look within the nodes to generate CMOCs at this stage, or were these already developed earlier in the process? It may help to include a diagram or add a column to Table 1 to show how CMOCs relate to the themes/nodes presented. I was also unsure why there appeared to be different types of nodes (e.g. sometimes seeming to refer to only contexts or outcomes and sometimes groupings of Cs, Ms and Os. Elaborating on details such as these would help the reader to follow the process adopted and increase transparency over methods. More detail is required on the methods of stakeholder engagement e.g. did consultations take the form of group or one to one discussion, how many people in total were involved from the different groups? Did stakeholders comment on the initial PT diagram, or one which was updated in light of learning from stage 2? I found the results to be very well presented and interesting to read. It is helpful that findings are traced back to the individual studies and CMOCs which support them. I did wonder whether integrating the findings from stakeholder consultations throughout the section on context-mechanism-outcome configurations might give a clearer sense of how these relate to aspects of the programme theory developed.
--	--

	The section on resultant programme theory is useful in showing how the initial PT was refined in light of subsequent review phases. As minor points for consideration, the meaning of abbreviations and symbols in Figure 2 are not always defined e.g. © and f/u Some mechanisms also appear to be missing from the table in supplementary file 6
--	---

VERSION 1 – AUTHOR RESPONSE

Reviewers' Comments to Author:

Reviewer: 1

Reviewer Name: Jean Macq

Institution and Country: UCLouvain, Belgium

Please state any competing interests or state 'None declared': no competing interest

Thanks for inviting me to review that interesting article. It presents very well the whole process of doing a good realist review. Methodology is globally well explained and how this translate into results is also thoroughly detailed.

As a consequence of those detail, it gives difficulty for the reader difficulty to “grasp” the main aspects of the theory, and the findings leading to it (i.e. one has to wait the conclusion to understand key points).

So, in order to make the paper more readable,

1) I would suggest to make “synthetic” points at different stage. Namely the initial PT and the updated PT lack of “clarity at this stage. On P4 line 49 the initial PT is presented on fig 2. It may be usefull to explain synthetically the content of that figure. Indeed, as it is, it is already very “complicated” (a lot of content). As a consequence, the reader has difficulties in the getting the “essence” of it, and the rationale of the initial PT. The same reader will therefore come with series of questions in the choices done in designing the initial PT (for example, why providers characteristics, interactions between GPs and hospital providers, ... is not mentioned in the context, and not taken in the mechanisms?).

We agree that step 1 should include an explanation of the content of the initial programme theory (Figure 2). Consequently, we have inserted the below section just before Figure 2.

Pages 4-5 “The initial PT shows two main channels for discharge communication; patient copied into (or not) the hospital to GP letter and patient received a personalised letter. Limited evidence was available for the option of ‘patient does not receive copy’ and this is evident in Figure 2. Patients being copied into discharge letters, whether by choice or otherwise, is associated with a large range of mechanisms and outcomes. Contexts such as ‘patient literacy level’ are likely to influence generation of mechanisms (patient does/does not find letter comprehensible) and outcomes (increase/no increase in patient knowledge), but this was unclear from the evidence reviewed in the scoping search. The scoping search revealed a range of “contextual influences” (e.g. ‘time constraints of writer’ may affect whether a patient is given a choice about receiving a letter and also the overall letter quality). It was unclear where some CMOCs began and ended (e.g. “is patient given choice about receiving copy of discharge communication” falls between contextual influences and context labels). In Figure 2, f/u stands for “follow up” and the small circled “c” icon refers to an outcome which could also thereafter take the form of a context. In summary, there were clear “gaps” and information missing from the initial PT, confirming that the scoping search in isolation was insufficient for realist

theory generation; further evidence and searching was needed to clarify details and simplify CMOCs (steps 2-6).”

On p15. As for the fig 2, it would increase visibility to explain synthetically the content of fig 4. We agree that a fuller explanation of the resultant programme theory (Figure 4) would be helpful. Consequently, we have inserted and re-drafted the below section to pages 15-16 which explains the content of figure 4:

Resultant programme theory

“The PT was systematically updated to produce a resultant PT following review steps 1-6 (Figure 4). This still shows two main channels for CMOCs: patient copies of letters and patient personalised letters. There remained limited CMOCs for where patients do not receive letters, due to the paucity of evidence available. Contexts for when the patient does receive their letter(s) were condensed into an aligned grouping of five key contexts for when the intervention may be theorised to work and four key contexts for when the intervention may be theorised not to work. The feasibility of providing a personal patient letter was updated on the PT; findings from both stakeholder feedback and data synthesis suggested personalised letters may currently be more feasible in private or insurance-based healthcare settings than in the NHS. In addition, the box of contextual influences was deleted, and the points integrated into the overall diagram. Patient outcomes throughout the PT were simplified and clarified (e.g. the outcomes such as ‘empowered patients’ and ‘reduce patient anxiety’ were simplified to the outcome ‘improved patient well-being’).

In addition, CMOCs have been “grouped” where overlap was apparent, for example, all resources are labelled simply as “resources” as data often concurrently referred to financial, administrative and clinician time resources. “Patient choice” emerges as a key influencer to the success (or not) of the intervention. Notably, there are a range of contexts, mechanisms and outcomes for when the intervention is theorised to “work” (e.g. positive outcome=improve doctor-patient relationship), and when it does “not work” (e.g. negative outcome=harm to patient). It is also evident that contexts can be used in combination to increase the likelihood of beneficial outcomes; this is indicated through wide arrows to show multiple linkages and amalgamations (e.g. a letter could contain no value judgements of patient and be written in simple plain English). Notably, the resultant PT has a higher incidence of CMOCs for when the intervention “does work”; this is considered further in the discussion.”

2) I would suggest to reduce details of some of the contents (for example, simplify the explanation on quality of evidence p9)

We have reduced and simplified this section of the manuscript:

“The findings were considered in light of the quality of included documents. During data extraction, documents were quality appraised for rigour and evaluated for relevance (22, 23). The concept of rigour is defined as ‘whether the methods used to generate the relevant data are credible and trustworthy’ (21). Relevance and rigour were scored on a scale from very low to very high and factors such as document type (e.g. opinion piece or scientific trial paper) were considered. Documents were not excluded solely based on rigour as extracts of documents with a lower quality score may still have valid contributions (23). The quality of evidence varied, with 53% of sources graded as medium or above for relevance and 80% for rigour. Information relating to setting and context was often limited.”

Few detailed comments

P3 line 53. Intervention is explained but not outcome. I would insert the explanation given on p4 line 16 just after the explanation of the intervention.

This has been changed.

P4 line 26. “what are the important contexts”.. add characteristics / components?... “which are associated..”

We are using “context” in our manuscript in the realist sense of the word. Context is defined in the way we use it throughout this review on page 3:

"Context may be conceptualised as external factors that influence mechanisms (22). "Mechanisms" are hidden, context sensitive causal forces that produce "outcomes".(22)

As such, we do not feel that there is a need to include additional detail within RQ2 as the realist concept of "context" already includes 'anything' that influences or triggers mechanisms (within CMOCs).

P7 line 45. Typo error? The word Intervention repeated twice
The second reference to "intervention" has been removed.

Reviewer: 2

Reviewer Name: Prof Maggie Bartlett

Institution and Country: University of Dundee School of Medicine, Scotland, UK

Please state any competing interests or state 'None declared': none

This is a well conceived, well planned and well written paper which follows the protocol for a realist review.

The background is adequately explained.

The methods section is very detailed and includes most of the necessary information, though the question of ethics approval needs to be addressed. Patients and NHS clinicians were involved and I would have expected the inclusion of a statement regarding the decisions about the need for ethics approval and their outcomes.

Details regarding ethical approval were stated in the published protocol but we agree should be re-stated here. We have added further details to step 6 to clarify. A new section has also been added and sub-titled "patient and public involvement" in accordance with the editorial comments at the end of the decision letter:

(Pages 8-9)

Programme theory refinement (step 6): Patient and public involvement

"Review step 6 was to consider stakeholder perspectives to test and refine the PT in light of the synthesised data (20) and to assess whether the PT aligns with real-life experiences (18). We invited comment from local policy makers and health service commissioners, GPs and a patient and public participation group. Groups were selected according to convenience and accessibility through University links. They were invited to suggest refinements to the PT in an entirely voluntary format, and all comments were anonymised. Formal ethical approval was not required (23) but informed involvement was sought."

The results section is clearly written and presented. The diagrams of the Programme Theory and its subsequent revision are comprehensive and easy to follow.

Reviewer: 3

Reviewer Name: Sarah Yardley

Institution and Country: Marie Curie Research Department, University College Hospital, United Kingdom

Please state any competing interests or state 'None declared': None declared

This is a really impressive and well conducted piece of work which only suffers from a few points of clarification and the need to more fully explain and justify elements of the methods and limitations. The authors should be congratulated on what they have achieved and with minor amendments I consider it to be a great addition to the literature.

My suggestions are as follows:

a) I accept the authors' point that benefit and harms that might arise from patients receiving or not receiving discharge letters has not been well researched or evidenced to date. However, it is quite hard to follow their logic in the abstract / introduction / methods (much clearer in the results but the reader needs early orientation) regarding the implications of this observation for the current review study as to how lack of understanding played out with respect to what outcomes were sought or not (including evidence about 'absence of outcomes' when constructing their programme theory. We have revised the abstract and made a number of changes throughout the document. Changes have been highlighted on the marked manuscript.

It concerns me however that as presented the study initially appears to have only looked for positive outcomes (see methods para 2 re the protocol) and how these could be reproduced rather than also looking for negative outcomes and/or unanticipated / unintended consequences (which might be either negative or positive).

We have clarified, and made additions where this applies, that we were looking for positive and negative outcomes throughout the review process. The review team felt "positive outcomes" such as increased understanding were as important to include in the theory as "negative outcomes" such as increased anxiety. The intended purpose was to develop a theory which explained how to increase the likelihood of benefits and reduce potential harms. However, as considered on page 18, more evidence was found for positive than negative outcomes. We have also added the following on negative outcomes to page 17:

"Important contexts for negative outcomes include: patient choice not acknowledged (56), only verbal information provided (44), letter involving terms and a style that is too advanced for patient to comprehend (78), and letter sent without verifying patient contact details (36, 54, 59)."

Later under review design negative outcomes are mentioned but it is still not clear if and how these would be accounted for in the analysis. The research questions then clearly state both negative and positive will be sought (RQ1) and contexts for both will be sought (RQ2) but again refer to whether mechanisms will produce different outcomes without specifying if this is all possible outcomes (as how defined e.g. in literature, by patients / stakeholders / by authors) or only those perceived as positive (and by who). Aside from referencing the protocol I think it is important this is discussed more clearly in the current submission, particularly as outcomes will not be mutually exclusive / binary in nature (a patient may experience 2 positive and 1 negative outcome for example) and it may be there is a programme theory to be developed for further work investigating if the counter to CMOs for negative outcomes would in fact be beneficial.

As described throughout the methods and protocol, CMOCS were not limited to one per document or one per patient experience and thus overlap is seen between CMOCs and positive and negative overlap within texts was accounted for with annotations. All labels for CMOCs and whether they were assigned to be positive or negative can be seen in Supplementary file 5. We were unable to include the table in the manuscript due to table size exceeding journal limitations.

To clarify who made the judgement regarding whether outcomes were labelled "positive" or "negative", we have added the following on pages 8-9:

"After table completion, following Pawson's framework (20), it was important to make sense of the "winners" and "losers". CMOCs were primarily labelled according to how evidence was reported in the included documents, such as whether the outcomes were described as desirable or beneficial. Where evidence was limited or outcomes were not clearly described or evaluated, the research team interpreted what data were available and formed judgements about these CMOCs based on content expertise in order to generate "positive" and "negative" labels. CMOCs were not limited to one per document or one per patient experience. Thus, multiple outcomes and CMOCs could be annotated for a single experience; this exemplifies the complexity of the intervention under scrutiny.

Notably, there were a greater number of CMOCs relating to positive outcomes than when the intervention does not work."

We agree that there is unlikely to be a binary distinction between positive and negative outcomes, and we have made this explicit in a new section added to the limitations on page 18:

“Additionally, the binary distinctions between positive and negative outcomes, that is, when the intervention does and does not work, may have imposed oversimplified CMOC labels. “Positive” and “negative” labels were based on evidence presented in the documents reviewed; at times a degree of subjectivity was involved in this process. Although these binary labels (positive/negative) may have oversimplified some CMOCs, we felt the usefulness of clear distinctions between when the intervention was interpreted to “work” (and not) outweighed the drawbacks of this method.” Changes and additions have also been made as follows:

Page 3

“The aim of the review is to understand and explain how the different outcomes are produced for adult patients receiving written discharge letters. Outcomes may be simplified into desired/beneficial or ‘positive’ (e.g. increased satisfaction) and undesired/detrimental or ‘negative’ (e.g. increased anxiety).”

Page 4

“Patients being copied into discharge letters, whether by choice or otherwise, is associated with a large range of mechanisms and outcomes. Contexts such as ‘patient literacy level’ are likely to influence generation of mechanisms (patient does/does not find letter comprehensible) and outcomes (increase/no increase in patient knowledge), but this was unclear from the evidence reviewed in the scoping search.”

Page 15

“It is also evident that contexts can be used in combination to increase the likelihood of beneficial outcomes; this is indicated through wide arrows to show multiple linkages and amalgamations (e.g. a letter could contain no value judgements of patient and be written in simple plain English). Notably, the resultant PT has a higher incidence of CMOCs for when the intervention “does work”; this is considered further in the discussion.”

b) I was surprised to see that only general practitioners and health service commissioners were included as stakeholders. The authors should consider the implications of not including hospital professionals who write discharge letters for both their findings and the potential impact of their study further in the limitations section.

Hospital professional views were considered throughout findings as these are evidenced in many of the included documents. It was not possible to include hospital professional stakeholders in step 6 due to lack of time. We recognise that this is a limitation, but as the review focussed on “receiving letters” we targeted groups most closely associated with letter receipt – patients and GPs in the first instance but also policy-makers. This has now been acknowledged in the limitations section as follows on page 18:

“Due to lack of time, it was not possible to involve all stakeholder groups who may be connected with discharge communication. Nonetheless, the review had a specific focus on “receiving” discharge communication and thus stakeholders were targeted who were closely associated or involved in policy of discharge letter receipt.”

c) Exclusion reasons

1. Why was mental health excluded? This needs better justification because if anything there is a history in Mental Health of greater transparency (not least due to the MHA as well as the MCA) and harm is often overestimated by professionals
2. Why were children excluded? While the letter would go to the patient's parents / guardian's for what reason did the authors anticipate this to be so different as to exclude?
3. It would have been, in my view, more appropriate to have included all letters related to discharge to the patient's home or usual place of residence e.g. other community teams rather than just GPs.

Can the authors comment further on these three decisions?

In all instances above I'm unconvinced by the justification that needs of these groups would be more complex and variable - surely this is a reason to include and not to make assumptions - inclusion

would have helped produce a more robust theory so this should be discussed as a limitation. In a small way not doing this reinforces the view of some healthcare professionals that their area of practice is a special case without evidence to support such a view.

As this review forms part of a PhD thesis, the review scope had to be contained within the limitations imposed on the PhD timeframe. The criteria for the review also aimed to mirror the criteria for the primary data elements of the PhD which went through NHS ethics but as the results are not yet published, this is not referenced here.

We acknowledge in the limitations section the impact of the exclusion criteria and the uncertainty regarding whether the PT applies or would be useful for excluded groups (or not). These groups warrant research in their own right and this could be a future avenue for research; this is indicated in the suggestions for future research section (pages 19-20).

d) I strongly advise the editors that table 6 is moved from supplementary data to the main article as this will strengthen the impact.

The review team agree with this suggestion. Please note this is now labelled supplementary file 5.

Reviewer: 4

Reviewer Name: Natalie Forster

Institution and Country: Northumbria University, UK

Please state any competing interests or state 'None declared': None declared

This paper makes an important and original contribution to current evidence by examining how, why copying patients into hospital discharge letters leads to different outcomes, and the key contexts which facilitate the success of this practice. The realist approach is well justified and appropriate to study aims. The programme theory developed gives a comprehensive picture of implementation pathways and outcomes (including potential unintended outcomes) of patient discharge communication. The review also draws out a number of useful suggestions for practice and further research on this topic.

However, I do feel that greater clarity could be provided on some aspects of study methods, particularly in regard to the processes of developing and refining programme theory. Specific comments are as follows: While the abstract gives an accurate summary of the paper, it would be helpful to add detail of the outcomes of receiving discharge letters that are under consideration in here.

We have revised the abstract which now reads as follows:

“Objective: To understand how different outcomes are achieved from adult patients receiving hospital discharge letters from inpatient and outpatient settings.

Design: Realist review conducted in six main steps: 1) Development of initial theory 2) Searching 3) Screening and selection 4) Data extraction and analysis 5) Data synthesis 6) Programme theory (PT) refinement.

Eligibility criteria: Documents reporting evidence that met criteria for relevance to the PT. Documents relating solely to mental health or children aged <18yrs were excluded.

Analysis: Data were extracted and analysed using a realist logic of analysis. Texts were coded for concepts relating to context, mechanism, outcome configurations (CMOCs) for the intervention of patients receiving discharge letters. All outcomes were considered. Based on evidence and our judgement, CMOCs were labelled “positive” or “negative” in order to clearly distinguish between contexts where the intervention does and does not work.

Results: 3113 documents were screened and 103 were included. Stakeholders contributed to refining the PT in step 6. The final PT included 48 CMOCs for how outcomes are affected by patients receiving discharge letters. “Patient choice” emerged as a key influencer to the success (or not) of the intervention. Important contexts were identified for both “positive” CMOCs (e.g. no new information in

letter) and “negative” CMOs (e.g. letter sent without verifying patient contact details). Two key findings were that patient understanding is possibly greater than clinicians perceive, and that patients tend to express strong preference for receiving letters. Clinician concerns emerged as a barrier to wider sharing of discharge letters with patients, which may need to be addressed through organisational policies and direction.

Conclusions: This review forms a starting point for explaining outcomes associated with whether or not patients receive discharge letters. It suggests several ways in which current processes might be modified to support improved practice and patient experience.”

The large amount of literature synthesised forms a key strength of the review. However, I felt the methods section would benefit from a more precise account of what the different review stages entailed, and how these fed into the development of initial programme theory. This would enhance transparency in regard to how findings were produced.

We agree that the methods sections presented here do not cover the full detail of the steps. We have already published the realist review protocol in BMJ Open which is referenced in the manuscript for this review which has a specific focus on the methodological steps. We have edited the methods section to make this clearer.

It is suggested that stage 1 involved a scoping search to locate existing theories, which were then used to develop an initial programme theory. It would be helpful to state what type of theories and evidence sources that were sought in this phase e.g. was this a search for broader abstract theory, or for existing articulations of programme theories in terms of how discharge letters are supposed to work? It is recognised that judgement is required in regard to the stopping point for developing programme theory, however it would be helpful to say how and when this judgement was reached, especially since this can be a common challenge when using realist approaches. How were decisions made to include articles in the scoping stage from among those available? A clearer distinction could also be made between the scoping search for existing theories (step 1) and electronic searches of the literature (step 2). The search terms/approaches seem very similar for these steps, yet within realist approaches, different search methods are often used depending on the level of theory being sought and developed (see for example Andrew Booth’s chapter in *Doing Realist Research*, 2018). A discussion of how these two stages differed from one another and their relative contributions to the development and refinement of the programme theory would be helpful.

We have addressed these comments through a re-write and multiple additions to the programme theory development (step 1). Further details regarding specific differences between the search strategies are found in supplementary files 1-3; these demonstrate that the scoping search was a quick simple search to grasp initial concepts and the full search was intended to be a more targeted search of evidence needed to develop the PT. A further discussion of the methods of the different searching steps is found in the published realist review protocol referenced in the manuscript. The new section for step 1 (pages 4-5) is below:

“Programme theory development (step 1)

The task of locating existing theories to develop an initial rough PT was achieved through a scoping search. Theories and evidence were sought which aided explanation of how and why patients receiving discharge communication results in different positive effects (e.g. drug adherence) and negative effects (e.g. preventable hospital readmissions). Sources were selected based on their “relevance” (19-21) to the PT; where relevance concerns ‘does the [source] address the theory under test?’ (20). Crucially, the whole source did not need to inform the PT but we considered the relevance and contribution of sections of the document (7).

Search terms were based on the intervention (e.g. patient cop(y)ies). Published resources and healthcare websites were searched to ascertain a range of evidence (see Supplementary file 1). During this phase, research team judgement was needed to decide the stopping point for programme theory development as was the need to balance the degree of comprehensiveness and practicalities

(23). As the purpose was to locate existing theories and initial concepts, the search was not intended to be comprehensive and the decision was made to screen no more than 30 documents. During the scoping search, search strategies within articles and article indexing were noted in order to inform a more thorough subsequent search in step 2.

Twenty seven documents were selected from the scoping search (see Supplementary file 2). All documents were then interrogated and coded for any CMOCs, concepts, or theories which could inform development of a PT. These were consolidated to form Figure 2, the initial PT.

The initial PT shows two main channels for discharge communication; patient copied into (or not) the hospital to GP letter and patient received a personalised letter. Limited evidence was available for the option of 'patient does not receive copy' as evident in Figure 2. Patients being copied into discharge letters, whether by choice or otherwise, is associated with a large range of mechanisms and outcomes. Contexts such as 'patient literacy level' are likely to influence generation of mechanisms (patient does/does not find letter comprehensible) and outcomes (increase/no increase in patient knowledge), but this was unclear from the evidence available in the scoping search. The scoping search revealed a range of "contextual influences" (e.g. 'time constraints of writer' may affect whether a patient is given a choice about receiving a letter and also the overall letter quality). It was unclear where some CMOCs began and ended (e.g. "is patient given choice about receiving copy of discharge communication" falls between contextual influences and context labels). In Figure 2, f/u stands for "follow up" and the small circled "c" icon refers to an outcome which could also thereafter take the form of a context. In summary, there were clear "gaps" and information missing from the initial PT, confirming that the scoping search in isolation was insufficient for realist theory generation; further evidence and searching was needed to clarify details and simplify CMOCs (steps 2-6)."

We have also added a few words to the introduction to step 2 to acknowledge the contribution of step 1 to the development of step 2 (page 5):

"The electronic searching was purposive and guided by the initial PT, results and indexing from step 1."

At times I found it difficult to follow the reporting of the synthesis process. I feel that a stronger thread is needed to show how the analysis progressed through the different stages and how these fed into one another e.g. how exactly did the authors move from annotations on individual articles (according to CMOC and PT ideas) to nodes? And then from the nodes to the CMOC table? E.g. did authors look within the nodes to generate CMOCs at this stage, or were these already developed earlier in the process? It may help to include a diagram or add a column to Table 1 to show how CMOCs relate to the themes/nodes presented. I was also unsure why there appeared to be different types of nodes (e.g. sometimes seeming to refer to only contexts or outcomes and sometimes groupings of Cs, Ms and Os. Elaborating on details such as these would help the reader to follow the process adopted and increase transparency over methods.

We have added new text to pages 7 and 8-9 as follows:

"Data synthesis using the analytic strategy 'juxtaposition of data sources' was achieved through utilisation of NVivo 'nodes'. Sections of text were annotated, and coded as nodes. The nodes were named according to ideas or concepts around the programme theory and contained sections of text that were used to build CMOCs."

"During, 'reconciliation of data discrepancies' (22, 23, 26) and 'adjudication of data' (22, 23, 26), the data coded within NVivo was used for scanning and comparing data to identify disparities.

Adjudicating and situating evidence was important to reconcile discrepancies (22, 23, 26). We interpreted the data coded within each node and judgements were formed as to which sections of text might be functioning as contexts, mechanisms or outcomes. We then made assessments about what the CMOC might plausibly be for each CMO based on the data within each node. Where relevant, we also drew on data contained within other nodes to build CMOCs. Following this process, a CMOC table was constructed (see Supplementary file 5) for consolidation of data.

After table completion, following Pawson's framework (23), it was important to make sense of the "winners" and "losers". CMOCs were primarily labelled according to how evidence was reported in the included documents, such as whether the outcomes were described as desirable or beneficial. Where evidence was limited or outcomes were not clearly described or evaluated, the research team interpreted what data were available and formed judgements about these CMOCs based on content expertise in order to generate "positive" and "negative" labels. CMOCs were not limited to one per document or one per patient experience. Thus, multiple outcomes and CMOCs could be annotated for a single experience; this exemplifies the complexity of the intervention under scrutiny. Notably, there were a greater number of CMOCs relating to positive outcomes than when the intervention does not work"

Mechanism information was limited at times and so some nodes do appear to form only arounds Cs and Os and others CMOs. Difficulties with mechanism evidence and the need to draw on team expertise for inference and stakeholder knowledge is acknowledged throughout the review. The results section is presented with the "nodes" as sub-headings and exact CMOCs from the table are referenced by number throughout this section. To clarify this we have stated the following on page 10: "The sub-headed themes emerged during data interrogation and consolidation although many acted as "nodes" in earlier annotation and coding (see Table 2)."

More detail is required on the methods of stakeholder engagement e.g. did consultations take the form of group or one to one discussion, how many people in total were involved from the different groups? Did stakeholders comment on the initial PT diagram, or one which was updated in light of learning from stage 2?

We have added an additional statement relating to the methods of stakeholder involvement on page 14-15 (also copied below). It is not possible to add further detail in order to protect the identities of the stakeholders. Moreover, exact stakeholder figures are unknown as contributions in group settings took place on a volunteer basis and therefore not all present necessarily took part. Furthermore, concepts and ideas often emerged out of lively group discussion and hence could not necessarily be traced to individual input.

Page 9:

"Review step 6 was to consider stakeholder perspectives to test and refine the PT in light of the synthesised data (23) and to assess whether the PT aligns with real-life experiences (21). We invited comment from local policy makers and health service commissioners, GPs and a patient and public participation group. Groups were selected according to convenience and accessibility through University links. They were invited to suggest refinements to the PT in an entirely voluntary format, and all comments were anonymised. Formal ethical approval was not required (18) but informed involvement was sought."

Page 14-15

"As detailed in step 6, the final review step was to refine the programme theory through stakeholder perspectives. Three groups were consulted: local commissioners, GPs and service-users/patients. Stakeholder involvement took the form of group discussions and email correspondence. As the PT was continually being developed throughout the review process, stakeholders commented on the most recently developed PT at the time of their involvement. Groups were relatively small; due to feasibility it was not possible to achieve diverse and representative group samples. Group discussions were centred on the programme theory; members were encouraged to critique and feedback on the PT diagram."

I found the results to be very well presented and interesting to read. It is helpful that findings are traced back to the individual studies and CMOCs which support them. I did wonder whether integrating the findings from stakeholder consultations throughout the section on context-mechanism-outcome configurations might give a clearer sense of how these relate to aspects of the programme theory developed.

Thank you for your comment. We have decided to leave the structuring in the current format in order to specifically highlight the contributions that stakeholders made to CMOCs where evidence was thin. This exemplifies the value of involving stakeholders in review work and we feel is one of the strengths of this review.

The section on resultant programme theory is useful in showing how the initial PT was refined in light of subsequent review phases. As minor points for consideration, the meaning of abbreviations and symbols in Figure 2 are not always defined e.g. © and f/u

We have added the following explanation to page 5,

“In Figure 2, f/u stands for “follow up” and the small circled “c” icon refers to an outcome which could also thereafter take the form of a context.”

Some mechanisms also appear to be missing from the table in supplementary file 6.

This supplementary file is now file 5. These gaps have left intentionally for transparency of where there was insufficient evidence to infer mechanisms. To clarify this, we have added the following sentence to limitations page 18/19:

“Additionally, not all mechanisms could be inferred from the data resulting in some visible mechanism “gaps” within the CMOC table (supplementary file 5).”

VERSION 2 – REVIEW

REVIEWER	Sarah Yardley Marie Curie Research Department, University College London, UK
REVIEW RETURNED	27-Feb-2019

GENERAL COMMENTS	I have reviewed the authors responses to all previous comments and feedback from myself and the other peer reviewers. I now recommend proceeding to publication.
--

REVIEWER	Natalie Forster Senior Research Assistant, Northumbria University, UK
REVIEW RETURNED	11-Mar-2019

GENERAL COMMENTS	I am satisfied that reviewer comments on this manuscript have been sufficiently addressed and I am therefore happy to recommend the article for acceptance in BMJ Open. As one minor editorial comment, I suggest that the use of the term 'hard-to-reach' on two occasions in the manuscript should be changed to an alternative (e.g. marginalised/disadvantaged/underserved) which does not blame groups themselves for a lack of access or uptake.
---